# HARP: Hierarchical Attention Oriented Region-Based Processing for High-Performance Computation in Vision Sensor

**DOI:** 10.3390/s21051757

**Published:** 2021-03-04

**Authors:** Pankaj Bhowmik, Md Jubaer Hossain Pantho, Christophe Bobda

**Affiliations:** Electrical and Computer Engineering Department, University of Florida, Gainesville, FL 32603, USA; pankajbhowmik@ufl.edu (P.B.); mpantho@ufl.edu (M.J.H.P.)

**Keywords:** computation at sensor, CNN, computer vision, image relevance, FPGA, ASIC

## Abstract

Cameras are widely adopted for high image quality with the rapid advancement of complementary metal-oxide-semiconductor (CMOS) image sensors while offloading vision applications’ computation to the cloud. It raises concern for time-critical applications such as autonomous driving, surveillance, and defense systems since moving pixels from the sensor’s focal plane are expensive. This paper presents a hardware architecture for smart cameras that understands the salient regions from an image frame and then performs high-level inference computation for sensor-level information creation instead of transporting raw pixels. A visual attention-oriented computational strategy helps to filter a significant amount of redundant spatiotemporal data collected at the focal plane. A computationally expensive learning model is then applied to the interesting regions of the image. The hierarchical processing in the pixels’ data path demonstrates a bottom-up architecture with massive parallelism and gives high throughput by exploiting the large bandwidth available at the image source. We prototype the model in field-programmable gate array (FPGA) and application-specific integrated circuit (ASIC) for integrating with a pixel-parallel image sensor. The experiment results show that our approach achieves significant speedup while in certain conditions exhibits up to 45% more energy efficiency with the attention-oriented processing. Although there is an area overhead for inheriting attention-oriented processing, the achieved performance based on energy consumption, latency, and memory utilization overcomes that limitation.

## 1. Introduction

The pixel-parallel CMOS image sensor has an escalating performance in providing a high frame rate with high-definition resolutions in video systems. This sensor has a signal processor for every pixel, used to process and store the pixel data, and provides a frame rate >10,000 frames/s [1,2]. Though the signal processors’ large footprint bounded the spatial resolution in pixel-parallel design, the 3D stacking architecture allows us to overcome this bottleneck by integrating pixel processors beneath each pixel [3]. These signal processors compute comparatively simple operations, such as extract temporal contrast in each pixel, low-level window-based imaging processing applications [4]. Here, the added benefits are massive bandwidth available at the sensor interface, and it enables us to meet the size, weight, and power (SWaP) constraints [5].

In contrast, high-level processing tasks like convolutional neural networks (CNN) require massive operations, and we initiate that processing at the cloud while the sensor works as a passive device. In that case, a camera collects pixels and then transfers every pixel to the cloud sequentially, which is shown in Figure 1a [6]. The pixel values propagate to the server through a channel that requires time *t_a_*. Here, both data dimensionality and propagation time increase with the increase of image size. The bandwidth constraint acts as a bottleneck for high-speed operations and creates a privacy concern. Some of the computations are performed at the sensor with a focal-plane sensor processor (FPSP) to improve the scenario [7]. The performance of the FPSP is limited to low-level functionalities. It increases the efficiency of edge-cloud collaboration. Edge-cloud collaborative computation is an excellent step to push computation at the edge [8]. Here, computational methodologies have been investigated where the signal processors in the sensor are assigned to perform more computationally expensive operations like convolution, and the host device performs the remaining function (see Figure 1b). Though the image size decreases after convolution, the number of channels increases, which further raises the image data volume. Therefore, the communication channel between edge and host still refrains us to exploit the maximum bandwidth available at the sensor interface.

This study suggests embedding an efficient inference engine to process image data at the edge device directly to collect useful information from the image (see Figure 1c). Since effective information in an image is minimal, the system demands a negligible data propagation time $t_{c}$. According to Figure 1d, $t_{c}<<t_{b}$ and it justifies that the data transportation time is no longer a bottleneck. This strategy further lessens the computational burden on the central node or host. However, the major challenge is that inference computation is costly and requires a large memory and computation power. A vast amount of parameters are associated with every learning method, presented in Figure 2. For instance, the VGG-16 learning model has 136 million parameters [9], and storing these parameters inside a vision chip is difficult. Aiming to mitigate all these problems, we propose hierarchical attention-oriented region-based processing or HARP. The HARP presents a hardware architecture to make an image sensor an active device by initiating hierarchical parallel computation at the focal plane. The hierarchical processing enables sensor-level information extraction instead of transferring raw data. Since the processing begins at the focal plane, there is high bandwidth for computation, making HARP a suitable candidate for high-speed applications.

Alternately, a real-time video system creates a data deluge with enormous redundant information, and the system spends unnecessary time and power executing the redundant spatiotemporal information. The retina in our vision system receives billions of information, but less than 1% data propagates to the central cortex. The retina performs some preprocessing to determine the regions where the visual attention implies. Inheriting these biological vision systems in vision applications provides the firmest connection to the low-dimensional fixational space and high-dimensional features or object space [10,11]. Hence, computationally expensive operations can be triggered by lightweight algorithms in the circuit to emulate vision systems [11]. This concept is tailored in the proposed HARP architecture by dividing the image into small image patches and then investigating the relevance of the image patches or regions. Figure 1e better illustrates this concept. The image is split into sixteen regions, where four regions are identified as interesting from our visual observations. HARP identifies these four regions with low-level image processing algorithms and triggers these four regions for high-level operations. Therefore, we can save power and increase throughput by enabling computation on the relevant regions only and gives an optimum platform to accelerate vision applications.

The state-of-the-art works for salient point extraction involved complex computation like color, orientation, and intensity computation in generating saliency map [12]. Determining spatiotemporal saliency is one of the common approaches to identify the relevance of a region [13]. HARP utilizes some low-level image processing algorithms (edge, corner, temporal contrast filter, predictive coding [14]) instead of complicated calculations. Our approach’s difference is that HARP does not generate a saliency map; instead, it investigates a region’s possibility to be marked as relevant with a series of low-level image processing. The major contributions of this paper are:We propose HARP architecture to perform low-level image processing and high-level inference computation at the image sensor, making sensor-level information creation.HARP introduces attention-oriented processing in the circuit to prune redundant spatiotemporal information and enables inference computation with a reduced amount of relevant data.Our model breaks the conventional nature of sequential image processing and applies massive parallelism to obtain high throughput by exploiting the large bandwidth available at the image source.The proposed method mitigates the high on-chip memory requirement and lessens the data movement among different layers. Furthermore, attention-oriented processing reduces dynamic power consumption and latency.FPGA and ASIC prototypes of our architecture are analyzed with different situations to observe the possible power and speedup.

The rest of the paper is organized as follows. Section 2 introduces previous works relevant to our approach. We present our hierarchical attention-oriented processing architecture in Section 3. Evaluation procedures, performance analysis results, and discussion are combined in Section 4. Finally, Section 5 concludes the paper.

## 2. Related Work

We organize this section by describing different hardware architectures for accelerating machine learning algorithms. GPU (Graphics Processing Unit) implementation, as well as custom hardware implemented on FPGA (Field-Programmable Gate Array) and ASIC (Application-Specific Integrated Circuit), have been explored to visualize the distinction in this work [15,16,17].

GPUs are a popular choice for training and testing a learning model because they give high speedup than Intel x86 architecture and excellent scalability in a large array format. Since GPU is a power-hungry device, we need an alternate option than GPU [18] for edge applications. Embedded GPU overcomes the drawback of high power demand, but it is not suitable for near sensor integration. Here, significant time is designated for data transferring, and it acts as a bottleneck. On the other hand, FPGA accelerators show better performance than GPU when latency and power are critical constraints. For instance, Spagnolo et al. proposed an energy-efficient hardware accelerator for CNN using heterogeneous FPGA. Their system on chip (SoC) architecture is structured to support the efficient Single-Instruction-Multiple-Data (SIMD) paradigm for computing both convolutional and fully connected layers [19]. Since all computations are applied on the FPGA and controlled by an embedded processor, they obtained better performance than the GPU. However, UltraScale FPGAs give more space putting more logic, and there are more research works in this direction to exploit UltraScale features. Authors in [20] used Kintex UltraScale FPGA to increase the maximum frequency and speed of the design by increasing the array of the convolutional processing elements. Inference computations are heavy operations, and authors in [21] emphasized optimization techniques to improve the scenario. They proposed an acceleration scheme for loop optimization with efficient dataflow and achieved 348 GOPs in Intel Stratix V FPGA. These hardware accelerators’ immediate attention is essentially focused on keeping the multiply accumulation (MAC) units busy and increasing the MAC units’ utilization factor.

Alternately, custom hardware architectures in ASIC overcomes some of the limitations in FPGA. ASICs are specifically designed for specific learning method or an application. They require a lower silicon footprint, lower latency, and power consumption than FPGA. These make the custom devices a better candidate for accelerating vision applications when low power consumption is demanded. Google designed Tensor Processing Unit (TPU) dedicated to accelerating machine learning models, and it is an ASIC equivalent multi-chip platform and a popular choice for CNN training [22]. Notably, TPU is more suitable in data centers rather than for edge applications. Alternately, Intel has introduced a neuromorphic chip Loihi [23]. It works with spiking neural network model with online learning capability, and the success of Loihi is yet to be manifested. Since FPGAs are power efficient, researchers are pushing them for edge computing [17,18,24]. Most of the hardware architectures follow a coarse-grained systolic array to implement accelerators and reads pixels from memory being at the edge. The systolic array type implementation gives a flexible design but exhibits high data movement with off-chip memory dependency. Furthermore, limited computation bandwidth is another hindrance to achieving high speed.

Hence, it is essential to overcome the limitations in data transportation time, and there are some works for better utilize the bandwidth available at the image source [5,25,26]. A programmable vision chip Scamp5d is proposed by Chen et al. which has SIMD-like parallel processing directly at the focal plane [25]. It has a 256 × 256 processing elements (PE) array for 256 × 256 pixels and a microcontroller to control the system. The PE includes light sensors, ALUs, registers, and local I/O buffers. This approach degrades the fill-factor of an image sensor. Alternatively, Bose et al. followed the same approach where using the Scamp5d demonstrated digit recognition using the MNIST dataset and car tracking in real-time [5]. The performance of the vision chip for large-scale datasets and CNNs is yet to be explored. It is challenging to have a complicated learning model in a vision chip when assigned at the edge for near sensor computation. Edge platform has limited resources with size, weight, and power constraints. Attention-driven strategy only shows the optimal path to reduce the power, high on-chip memory demand, and time. Event-based or selective change driven (SCD) cameras were introduced to demonstrate the attention in the form of temporal change [27,28]. These cameras transfer positive or negative events from a scene if there is a temporal change and do not carry the intensity information. An external server collects these events and associated addresses, performs image reconstruction, and then applies vision algorithms [29]. This process mitigates the data transportation latency between sensor and server, but event cameras’ effectiveness for time-critical vision applications is yet to be explored. Therefore, collecting the intensity in interesting regions and processing at the collection point is inevitable to achieve the best performance. Lee et al. had focus on the intensity with an attention-based strategy for removing redundant spatiotemporal information from an image [13]. Later, Samal et al. exploit this concept and proposed attention-based pruning from image to reduce the amount of active information that can reduce the data flow in the hardware [30] based on each pixel. It is possible to achieve more optimization by doing region-level analysis. We can make an entire region zero when a region has less relevance.

To the best of our knowledge, we are the first proposing HARP architecture where we leverage visual attention procedures with pixel-parallel CNN operation tailored with the region-based approach to push the overall computation directly at the image sensor’s focal plane.

## 3. Proposed Architecture

This section describes the proposed HARP architecture of the pixel-parallel CNN inference model at the sensor, illustrated in Figure 3. The HARP architecture demonstrates hierarchical processing in the readout circuit of a pixel-parallel image sensor implemented by Sony [2]. The execution procedure of HARP is split into two logical layers: Attention-Based Preprocessing Layer (APL) and Inference Computation Layer (ICL). The sensor performs readout after completing the hierarchical operations in these two layers. The APL works on an *M*×*N* image which is logically split into *M* smaller image patches or regions. Each region is identical and has a 2D array of *N* pixels. Here, these numbers are design choices, and we can increase the degree of parallelism or throughput by increasing the value of *M* or making smaller regions. The APL identifies the relevance of each region by early feature extraction. Next, the ICL computational layer has an array of region inference engines (RIE). Only the relevant regions are mapped into the RIEs. A fully connected neural network (FcNN) sequentially receives the output from each RIE. These hardware modules are combined with an embedded processor and create a system on chip (SoC) environment. Figure 4 is a simplified flowchart of the proposed HARP architecture. The detailed explanations of these logical layers are described in the following sections.

### 3.1. Attention-Based Preprocessing Layer (APL)

The hardware design of the APL is shown in Figure 3, which is the first design phase. This layer takes pixels from a pixel-parallel image sensor, (implemented by [2]) directly. It is a 2D array of *M* region processing units (RPUs) (see Figure 5a), and each RPU has *N* (256 in this implementation) pixel processing units (PPUs) with an Attention Module (AM) (Figure 5c). They collect pixels from the sensor in parallel, perform low-level operations to find image relevance. The detailed descriptions are presented below.

#### 3.1.1. Image Acquisition

The image acquisition process begins at the pixel level, and it is depicted in Figure 5b. The PPU array receives the pixel value in parallel directly from the image sensor [1,2]. Each PPU broadcasts the received pixels in its east (E), west (W), north (N), and south (S) direction. These four channels are used to transfer and receive neighboring pixels in a time-shared manner. So, center pixel C receives the four pixels from four channels. The operations assigned to the PPU also require the corner pixels in a 3 × 3 kernel. To get corner pixels, every PPU acts as an active forwarding unit. After receiving the direct channel pixel values, every PPU broadcasts its north and then south pixel. According to Figure 5b, when W broadcasts its north pixel NW, the center PPU C receives the corner pixel. Using this method, every PPU receives all neighboring pixels, and the acquisition time is irrelevant to image size.

#### 3.1.2. Lightweight Method of Relevant Region Detection

After the image acquisition process, every PPU in the array has the neighboring pixel information, and we perform a certain number of operations to identify spatiotemporal redundancies in the image. Every PPU executes in parallel, and its functionalities are divided into several modules.

Figure 5d depicts every module in the PPU. The M0 module has a pixel and memory register array to store the pixel values for a specified window at the time *t* and (*t* − *1*), respectively. If the comparator finds a notable change that is greater than a threshold in the window, then it gives one as output to module M3 and updates the memory register. The value one represents that the pixel has temporal saliency (TS). Alternately, the M1 module determines the spatial saliency (SS) associated with the pixel using the Sobel edge detection algorithm. According to Figure 5d, M1 has the convolutional kernel to determine whether the pixel lies on edge. M1 gives one as SS score if the pixel lies on edge and zero otherwise. Simultaneously, the M2 module performs image enhancement applying the Gaussian smoothing function. It helps to remove noise from an image.

We need to transfer SS, TS, and pixel values produced at the PPU for the subsequent processing. The output of the PPUs form a scan-chain shift register, and every clock cycle, the AM dedicated for every region receives the output of the PPU group (see Figure 5c). M3 module in Figure 5d enables this function. At the first clock cycle, the data produced in the PPU goes to the output, and the rest of the cycles, they only forward their data.

The AM, shown in Figure 5c, receives the SS and TS values and the region output buffer (ROB) receives the pixel values in a sequence. The ROB is an asynchronous first-in-first-out (FIFO) that maintains coherence between APL and ICL. The AM receives and accumulates the TS and SS scores from every PPU and generates two one-bit tags: regional temporal saliency (RTS) and regional spatial saliency (RSS). These tags are one if their corresponding accumulation is higher than a threshold. If both tags are one, then we call the region as a relevant region and irrelevant otherwise. The subsequent processing and output for that processing depend on these values. Table 1 presents the impact of RSS and RTS. Operation in the next layer for this region begins if and only if RSS and RTS are one. In this case, the processor gains control of its output buffer. Alternately, if one of the two values is zero, the corresponding processing is halted, and the output register value becomes dependent on the tag values. When the RSS is one, and RTS is zero, we do not update the register value. Because we calculate with the same set of pixels at the preceding time. Besides, when RSS is zero, RTS becomes a donot care state. The attention module makes the output zero and makes the region blank since it does not carry meaningful information. The main difference with the existing works is that we determine relevance in every small region to initiate region-based operations and obtain output in the irrelevant regions by skipping the execution. We do not involve computationally expensive operations for generating a saliency map.

We explore different datasets to estimate the benefit of the proposed method. The analysis for the MOT17-SD1 dataset is pictured in Figure 6 [31]. The dataset contains video clip images (1920 × 1080) of people walking around a large square. The image stream was captured with a 30 FPS camera. The dataset has 450 images arranged in sequence. We consider 8 × 8 region size to analyze these datasets and make the regions with temporal redundancy blank or zero. Notably, region size is a design choice. When the region size is large, the interesting regions may contain a significant amount of redundant information. In contrast, a smaller region size is effective in filtering redundant data but increases the hardware cost. Figure 6b represents the analysis of inter-frame temporal redundancies. In some cases, the redundancies are more than 90%, and on average 71% regions in the 450 images are redundant. Besides, we calculated the spatial redundancy on MNIST and FashionMNIST dataset. These datasets contain ten thousand images, and we also split each image into an 8 × 8 spatial region and apply the spatial saliency calculation methods. In each region, we collect the number of pixels with information and define spatial saliency. When a spatial saliency is less than a threshold, we prune a region. The pruned images for different threshold values are tested with the pre-trained learning model. This analysis has been tabulated in Table 2. We found that MNIST and FashionMNIST datasets have 50% and 29% regions are redundant on average. After pruning these regions, we did not concede any accuracy drop. Adding to it, Ghani et al. also demonstrated a region-of-interest-based image classification where they passed only the ROI data of an entire image for training a neural network and achieved 100% accuracy [32]. Therefore, it is an effective solution to prune redundant spatiotemporal information from images first and then apply the learning method to expedite the overall processing with optimal computation cost.

### 3.2. Inference Computation Layer (ICL)

Inference computation is the second phase of our design, and the hardware design for near sensor implementation is pictured in Figure 3. The submodules in ICL take a region if the region has relevance (see Figure 4). Then ICL initiates the CNN processing with a region-based approach using the data from a relevant region only.

According to Figure 3, the entire CNN operation is executed by a set of region inference engines (RIE) and a fully connected neural network (FcNN) module. An external embedded processor is connected with the RIEs and FcNN. The embedded processor has pre-trained weights organized in a defined pattern and gives an off-chip memory interface. The RIE overview is presented in Figure 7a, where each RIE has two processing modules. The first module is the first convolutional layer (FCL), and the second one is called the second convolutional layer (SCL). The FCL and SCL perform convolution, max-pooling (when required), and rectified linear (ReLu) operations in parallel. After the convolution operations, FcNN performs the execution to generate the output. The novelty here is that, among *M* regions, only relevant regions are mapped in the inference engines. In between the APL and ICL, there is a scheduler (Figure 3), and the scheduler is responsible for mapping the active regions into *K* inference engines. The scheduler mimics the concepts of semaphore locking. It first serializes the active regions and creates a *K*-bit key. The *K* regions out of all active regions in the image get the *K* inference engines and enable the lock. When any one of the inference engines finishes the execution, it releases the key, and another region in the pipeline gets access to the engine. The ICL has a tag register that receives every region’s saliency scores from the APL through the scheduler.

#### 3.2.1. Region Inference Engine (RIE) in the ICL

Here we describe how the RIE performs convolution operation, internal architecture, and its configurable datapath.

(a)Convolution Computation Process in each RIE:

The FCL and SCL are the two modules in an RIE, and they perform convolution, pooling (when required), rectification, and quantization operations.

An RIE works in a small region if it is relevant. In general, every convolutional layer in any learning model has multiple output channels. All these output channels increase the data dimension, and data volumes increase as a consequence. The increased volume is a serious issue because we need large on-chip memory to hold all the feature maps, which is difficult to manage for an edge device like an image sensor. HARP has FCL and SCL and jointly works on partial data-based computation to mitigate this problem. Lets assume the first convolution layer in a learning model has *p*×*q* kernel size with *r* output channels, where the second layer has the same kernel size with *s* output channels. In this case, the FCL generates one output channel with the region data, and then the SCL acquires that data and produces data for *s* channels. Here the data in the *s* channels represent an incomplete feature map, and we call it partial data. Next, the FCL gives another channel’s output, and the SCL makes another set of partial data and accumulates with the previous partial data (see Figure 4). This process iterates until every channel in the first convolution layer is executed. It is notable that the FCL always finishes the computation prior to the demand by the SCL. When FCL or SCL finishes their computation, two intermediate output buffers (IOB) collect their output data. An external memory fetches the data from the second IOB (see Figure 7a). When RIE finishes executing on the first two convolutional layers, then the FCL acquires data from this external memory to initiate processing in the third and fourth convolutional layers. According to Figure 4, this method continues until we process all layers.

This strategy saves the high on-chip memory requirement and executes every second convolution layer in a learning model without accessing memory. Generally, the first two convolutional layers are more expensive; for instance, the VGG-16 network requires 3.2 Mbytes [9], where most of the data is not relevant. Therefore, our image feature-based optimization method plays a vital role in reducing power and latency. The pairwise convolution operation further reduces energy cost by fetching a region data from the external memory and executes two convolutional layers in the FCL and SCL. Hence, the quantity of external memory access drops by 50%. The RIE has a configurable datapath that allows the SCL to read directly from the external memory (see Figure 7a). The configurability brings flexibility in the design because many learning models have odd-numbered layers (for example, VGG-16 learning model has 13 convolutional layers). In this case, the RIE computes pairwise convolutional operations, and finally, the last layer of convolution executes on the SCL.

Here, the external memory access does not cost time because the SCL execution time is larger than the FCL. FCL fetches data from the external memory while the SCL remains busy working with partial data. Thus, the design does not have any waiting time to fetch data from external memory. Adding to it, we merge the saliency score in the tag register’s content as we move up in the network. For instance, when the RIE works on the third and fourth convolutional layers, we redefine the saliency score. As we go deep in the hierarchy, region size reduces for convolution and pooling operations. Each iteration brings the same amount of data from the external memory to make the data compatible with the hardware resource. The data that FCL fetches represent more than one region in the original image. We redefine the saliency score by doing OR operation among the contributing region’s saliency scores. Based on the new score, the system determines the operation on that set.

(b)Architecture of the FCL and SCL:

The operations in the FCL and SCL are essential to understand the architecture. Their functionalities are similar, and Figure 7b represents the FCL/SCL block diagram.

The FCL or SCL is an array of pixel processing units (PPU) and processing elements (PE). The PPUs in the FCL receive data stream directly from the content of active RPUs for executing the first convolutional layer and then accesses the external memory to run the other layers. Alternately, the SCL mainly collects its input from the FCL. The number of PPUs in FCL or SCL is equal to the number of features they receive. However, for every spatially distributed 2 × 2 PPU unit, the design considers a PE. The reason is, every PPU finishes pixel acquisition in sequence, and we pipeline the processing in the PE to reduce the number of required multiply accumulation (MAC) units for pixel parallel processing. Furthermore, the four PPUs share their common pixels, and the mapping of these pixels in a single PE keeps hardware overhead low. Hence, it exhibits optimization for latency and area. The dataflow and the execution process are described below.

Image and weight propagation: Efficient image and weight propagation are vital to improving the throughput of the design. Here we discuss the steps we have taken to improve the scenario.

It is essential to have a fast data movement in the PPU/PE to increase the MAC units’ utilization. The first PPU in Figure 7b receives the image stream and transfers using two channels to populate the PPU in its next row and column simultaneously. All PPUs in the first column participate in the bidirectional distribution. Other PPUs, forward the stream only to its right at every clock. The bidirectional data transfer expedite the streaming process compared to the conventional system [6]. It needs (*g × h + g + h − 1*) clock cycles, where *g* and *h* are the numbers of column and rows of that region. Alternatively, to perform MAC operations, we also need to update weights prior to the demand. The weights in Figure 7b is a small memory unit, and it fetches the corresponding weight set from the global memory and remains updated before the execution demand.

Execution Process: Figure 8 shows the execution process of a PE. When the top-left PPU for a PE finishes acquisition (Figure 7b), then PE takes that data and initiates MAC operation in parallel on the given kernel size. After that, it loads the PPU data to its top-right direction in the following cycle, and in this fashion, it continues the execution. Next, rectified linear unit (ReLu) in Figure 8 receives the data after convolution and induces non-linearity by inducing non-negative suppression. Here, the benefit of ReLu operation is straightforward and exhibits less area overhead than popular sigmoid functions. The operands of the data in the convolutional units are 14-bit wide. To achieve more computational correctness, we make the intermediate signals signed 28-bits. These signals after the rectification are transferred to the quantization unit to convert 14-bit data. Besides, if the convolutional layer in a model requires the pooling operation, we can perform that operation before quantization and skip the function otherwise (Figure 8). The pooling unit is configurable to perform max-pooling, average pooling, or min-pooling. Hence, PE offers a flexible design in the execution pipeline. Besides, the PEs in the SCL performs some additional task. After the execution process, they do not transfer the data. Output register of each PE store the data locally and perform addition and accumulation to add the partial data.

Writeback operation: The process of writing back to the intermediate output buffer (IOB) is shown in Figure 7b. To achieve high-speed processing, it is also essential to collect the outputs and make the unit ready for the next processing. The output of the PE works as a scan-chain shift register. When output becomes ready, every PE sends their output to the output register, and that value propagates sequentially to the IOB. FCL writes to its IOB after computing one channel. However, SCL writes to its IOB after generating the complete data from the partial data. Notably, the output buffers are asynchronous FIFO.

(c)Configurable datapath in between the FCL and SCL:

The FCL and SCL have a configurable datapath in between to execute multiple convolutional layers. The FCL takes data from a region, executes that, and commits writeback to the IOB. When there is new content in the IOB, the SCL starts receiving the feature data to initiate the second convolution. The FCL or SCL can take or transfer data to the external memory when required (Figure 7a). Mapping flexibility is one of the design’s key features to adopt different learning models with different numbers of convolutional layers.

#### 3.2.2. Fully-Connected Neural-Network (FcNN)

A fully connected network is the last layer of the design, and it extracts information that is transferred outside of the sensor by the readout circuit (ROIC) when that information has relevance (see Figure 3). This layer is also close to the sensor. It is configurable to adjust with different neural network models. Figure 9a shows the functional diagram of the FcNN. Figure 9b represents a typical fully connected network which has P × Q nodes in layer-1. This figure suggests that FcNN has a large number of weights but a small number of operations, which makes the throughput for hardware primarily bounded by the off-chip communication speed. Our method pictured in Figure 9c mitigates the bottleneck.

A FIFO provides one data and memory provides *P* number of weights to the FcNN (see Figure 9a,c). The FcNN works like a systolic array of *P* MAC units. Layer-1 in Figure 9b has P × Q nodes that needs P × Q multiplications on a single data. As a result, the process iterates Q-times to complete executing each data. It is inefficient to keep the weights for every pixel in memory; rather, we store 2 × P × Q weights. It emulates the dual buffer technique and provides *P* weights to the MAC units at every clock cycle. The memory keeps updating the weights from the external memory to avoid stalling in MAC operations and using large on-chip memory. Alternately, the FIFO data contains a tag that represents spatiotemporal redundancy. If the data is redundant, then we skip the MAC operation for the pixel. Furthermore, we use the conventional approach of checking nonzero data to allow multiplication. In Figure 9c, each MAC unit has a register that holds a *Q* number of data to form P × Q nodes after executing layer-1. All registers also form a chain to shift their content. When layer-1 finishes its task, the register content feeds the MAC units and iterates until all multiplications in layer-2 are executed. We repeat this process to complete the task in the fully connected layers. Adding to it, these MAC units perform multiply accumulation, quantization, and rectification (Figure 9d).

We would like to note that, though we have an embedded processor in the architecture, the computation does not depend on that processor. It mainly maintains configurable communication signals, which does not degrade the performance. The processor works as an interface to apply the sensor for different learning models. In summary, the HARP uses hierarchical bottom-up processing, which begins at the pixel level. To save hardware cost for high-level inference processing at the image sensor, low-level preprocessing prunes the irrelevant regions allows a set of regions with where our visual system may fixate. The pixel-parallel operation with deep pipeline operations among the APL, RIE, and FcNN on the reduced amount of information makes the overall design running at comparatively low power and gives time savings while utilizing comparatively lower memory.

## 4. Result

We organize this section with the evaluation infrastructure, detailed implementation, experimental results, and performance analysis.

### 4.1. Evaluation Infrastructure

Maintaining the ASIC design flow, we develop the design specification and finally obtain a GDSII file with a physical layout for the HARP architecture. It is a block-level design implemented in 90 nm technology using the TSMC standard cell library. We describe behavioral description in the SystemVerilog and verify in Synopsys VCS tool. Synopsys Design Compiler gives a gate-level netlist, further verified by the equivalence checking through Synopsys Formality. Finally, we obtain the GDSII file by performing place and route in Cadence Innovus from the gate-level netlist and Synopsys design constraints. HARP is also implemented in the Virtex UltraScale+ FPGA board (xcvu9p-flgb2104-2-i) using Vivado design suite 20.1 for prototyping by maintaining the FPGA design flow.

### 4.2. Evaluation Metric

The HARP architecture can be evaluated for different classifiers and object detectors. As a case study, we have used the VGG-16 classifier, which is challenging to push for near sensor computation. Here, the image size is 224 × 224 and we divide that into 196 regions where each region size is 16 × 16. We have also considered the neighboring pixels of every region that makes the effective region size 18 × 18. According to HARP strategy, RPU in the APL has 16 × 16 PPU array, FCL and SCL in the inference engine have 16 × 16 and 14 × 14 PPU array, respectively. The FCL fetches 18 × 18 data for every region and generates an output of 16 × 16 if there is no pooling. Here, these 68 additional pixels are neighboring pixels of a region. We consider pre-trained weights in the external memory and maintain fixed-point 14-bit precision. All these parameters have also been tabulated in Table 3.

### 4.3. Implementation Details

The design has been prototyped in FPGA and ASIC using the aforementioned evaluation metric. We analyze latency, area, and energy parameters for evaluating this design. While designing, one of the objectives was to keep silicon footprint and power low as much as possible without compromising latency. Besides, maximizing the system frequency and minimizing the memory utilization are also our significant contributions to this implementation.

#### 4.3.1. Silicon Footprint Analysis

HARP architecture breaks the conventional sequential processing and introduces parallelism with the pixel-parallel operation, leading to higher area demand. The design stands to perform inference computation, but the APL layer serves a different purpose. Hence, the area associated with the APL design is considered as an overhead.

We implement the overall design according to the description given in Section 3 using our infrastructure. We obtain the optimized layout of this model in FPGA and ASIC, presented in Figure 10. The layout extracted parameters for FPGA and ASIC have been tabulated in Table 4 and Table 5, respectively. The estimated chip area with four inference engines is 8.34 mm${}^{2}$, and the APL layer is 11.34% compared to the overall design. When compared with a pixel circuit, this area gives only 2.78% overhead. In this case, the pixel circuit includes the photosensitive elements and an in-pixel analog to digital converter. We estimate the area based on the most optimized ADC in the 90 nm technology [33]. We sacrifice the extra 11.34% resource to save latency, energy, and memory. The area associated with the inference computation layer (ICL) represents four inference engines’ total area requirement plus the FcNN. As mentioned in Section 3 they work independently, and among the engines, there are no dependencies. Hence, based on the system requirement, it is possible to determine how many RIEs need to be integrated for a particular vision application.

#### 4.3.2. Analysis on Energy and Latency

Table 4 and Table 5 also include the power consumption associated with each module. The APL consumes 11.81% of the total power, which is an overhead. Nevertheless, the advantage is that HARP consumes only 11.81% power plus the leakage power by the inference engines in the irrelevant regions without consuming 100% power that we report. Hence, the dynamic power consumption stays low. Alternately, when the APL finishes reading an image frame, it goes into the idle state. Because the ICL layer needs more time to execute an image frame, and the data gets lost if that comes before they get ready. This gives the possibility to keep the APL inactive, and they make data available prior to the demand of new data by the inference engines in the ICL. We apply the clock gating method to reduce dynamic power consumption, where the enable signal to enable the clock comes from the attention modules. Using the power gating method, we can save leakage power but is it beyond the scope of this paper. Leakage power is around 2.3% (see Table 5) for most of the modules; we are not focusing on mitigating that.

We provide the time required to process a set of data by each module in Table 5. The delay is an important metric to evaluate the performance of the design. The APL layer delay is very low (0.710 µs), which is the only 17% of compared to the inference engine computation time. Without the APL layer, a system would have spent more than 4.1 µs on irrelevant regions to identify that there is no feature point. However, the APL layer predicts that 5.2× faster than the inference engines. On the other hand, the delay in the FcNN is 0.01 µs, which represents the latency for a single data. It is an array of 100 MAC units, and we iterate the computation to finish the required amount of multiplications in the fully connected layer. However, the learning models have multiple convolutional layers. These comparisons are made only for two computational layers to explain the benefits of our architecture. Unlike APL, energy and time increase in the inference engines with the rise in convolutional layers. Hence, APL’s importance is more prominent in a learning model of a classifier or an object detector, which has a lot of convolutional layers.

#### 4.3.3. Impact of Irrelevant Regions on Power and Execution Time

We have seen that the APL brings the advantage of saving power and time, and now we quantify the benefit for some images with different percentages of redundancies. Figure 11a represents the results with five images where the number of redundant regions varies from zero to a hundred percent. In the Y-axis, we present the performance of the HARP architecture with and without the APL layer. The HARP without the APL layer performs execution continuously, but with the increase of redundant regions, both execution time and latency go down. When there is 0% redundant information, this architecture with the APL layer draws more power and needs more time. However, we have discussed in Section 3.1.2 that the overall natural image has huge redundant information. Execution time drops by 70.1% with 25% relevant information when we apply attention-oriented processing. This processing also saves significant power, as illustrated in Figure 11a. For example, if every region is redundant, power consumption drops by 76.34%. As a result, the power consumption and latency are always lower when we consider attention-oriented hierarchical processing. Therefore, the APL integration in the HARP is crucial for accelerating vision applications in resource constraint edge devices.

#### 4.3.4. Pipeline Processing for Faster Operation

From the delay analysis in Table 5, we can visualize that the design spends more time in the inference engines. The delay in the ICL layer increases with the increase of convolutional layers and channels in each layer. Hence, an efficient pipeline is required to increase the processing speed. Figure 12 presents some of the simulation results to illustrate the pipeline processing, and Figure 11b demonstrates the clock cycle-based presentation of FCL and SCL. In Figure 11b, X-axis represents the FCL or SCL’s repeating operation, and the Y-axis shows that clock-cycle. An RPU takes 359 clock cycles to determine a region’s saliency and makes the load “true” for the ICL. When the load becomes true, the FCL starts fetching data from that region and executes the first output channel. When the first channel data is available then the SCL initiates execution (see Figure 11b and Figure 12). Time requirements associated with image fetch, execute, and writeback for FCL/SCL are presented in the inscribed Figure 11b. The initial pixel acquisition time for the FCL is longer and takes 359 clock cycles to populate the PEs, which is 1.81× faster than the conventional streaming process. The SCL also achieve this speedup to read 256 data and populate 196 PEs by 288 cycles. The FCL remains unused for a certain period of time, and we update weights in the weight-buffer for the MAC operations to make the design more efficient. In addition, we also illustrate in Figure 11b that the FCL always makes partial data ready to avoid stalling in the SCL. The SCL takes the output from FCL to produce a partial result and then performs writeback in the last iteration when the complete feature map becomes ready. Therefore, HARP architecture executes does not require accessing the off-chip memory to execute the second convolutional layer. Hence, this pipeline exhibits an additional acceleration by minimizing the dependency with the off-chip memory.

#### 4.3.5. Memory Optimization

The proposed architecture aims to reduce the memory overhead while doing the computation of a neural network. The VGG-16 has a very high memory requirement- it needs 96 Mbyte total memory [9]. It is very challenging for any small chip. The partial-data-based implementation strategy requires only 128.5 Kbyte memory for storing the features in every RIE output. It is 1.99× less than the on-chip memory requirement for computing two convolutional layers. Besides, the added benefit of two convolutional layers is that we reduce the amount of on-chip to off-chip communication.

#### 4.3.6. Maximum Frequency

We achieve high-performance by ensuring that the design runs at a high operating frequency. Both FPGA and ASIC prototypes have been taken into our consideration to maximize the frequency. The design is comparatively large, with many independent modules. Each layout in Figure 10 forms asynchronous boundaries in the SoC design and assigns a separate clock for every module. When a data faces the clock domain crossing, there is a high chance for metastability [34]. We avoid metastability by dual-port asynchronous FIFO. Furthermore, we use a synchronizer for the control signals among the modules [34]. We split the modules among the three super logic regions (SLRs) available in the Virtex UltraScale+ FPGA [35]. Then we use the built-in phase-locked loop (PLL) in the FPGA to maintain synchronization among the clock domains. Like ASIC, asynchronous FIFO and synchronizer prevent the metastability in the FPGA [34]. While designing these modules, we perform static timing analysis. We work in the critical paths and reduce the combinational logic delay to fix the setup time. All these steps help to avoid setup and hold time violations. Finally, we achieve 320 MHz in the FPGA and 430 MHz in ASIC.

### 4.4. Case Study on Classifier

We analyze the VGG-16 learning model as a case study with the ades and culez mosquito species image dataset [36]. The reason we choose the dataset is that it has higher redundant information with high image resolution. However, this HARP architecture’s scope is not limited to these classifications; instead, we can use this model for other real-time applications. We logically split the images into equal 196 regions where each region size is 16 × 16. The preprocessing for spatial saliency makes the 28% homogeneous irrelevant regions blank. We develop the software modeling of the HARP architecture with the VGG-16 learning model. The model uses transfer learning and is trained with the original and our modified datasets. Notably, there is no accuracy drop for pruning the irrelevant regions.

The VGG-16 learning model for the HARP has seventeen layers. The first layer is the APL layer, then thirteen convolution layers followed by three fully connected layers. Figure 13 summarizes energy consumption and latency associated with each layer. This figure presents three test conditions: (Case-1) all regions are relevant, (Case-2) 50% relevant information, and (Case-3) 10% relevant data. Here, case-1 is an overestimation, but case-2 and case-3 are common for a natural scene or real applications. According to this figure, HARP shows that case-1 exhibits the highest energy consumption and latency among the three cases in every layer. With the decrease of relevant regions, the energy and latency requirements drop sharply. We achieve improvement up to convolution layer 3-2 (which is the sixth convolutional layer in VGG-16), and after that, the improvement is not significant. The reason is that after so many convolutions, the image becomes small (28 × 28) and data becomes too sparse, and then we cannot eliminate a region from being processed unless 100% data is redundant. This situation has a small impact because the last four convolutional layers are not expensive (see Figure 13). When HARP discards a region, that gives huge benefit to save latency, it directly saves the memory access time to fetch data of a region. For instance, the first convolution layer is 15.82× faster when there are 10% relevant regions. We would like to note here that when every region is redundant, then the energy and latency will be close to zero in every layer. The APL remains active in this particular case and performs surveillance to obtain a possible object in the image.

Overall, we save 45.82% energy and expedite the system by 35.63% when we have 10% relevant information in the image. Hence, attention-oriented processing has a huge benefit.

### 4.5. Performance Comparison

This section summarizes some state-of-the-art competitors when executing the VGG-16 model or runs CNN at the edge platform.

We compare our work with GPU, FPGA accelerators, and some recent works where CNN is implemented at the sensor and tabulate in Table 6. The peak performance or computation roof of a hardware design is obtained by the maximum number of operations when every required data is available in on-chip [37]. The computational roof is hardly gained by a design because MAC units may remain underutilized for the dependency with external memory, data-dependency, or memory contention. While we are making the performance comparison, these issues were taken in our consideration. GPU popular choice for CNN operation. VGG-16 is a vast network, and a powerful GPU like GTX Titan Black achieves 7.8 frames per second [9]. Here, HARP gives **3.68**× better performance than the state-of-the-art of GPU implementation. The table also includes the comparison with some of the novel FPGA accelerators. The motivation of the authors in [19,20,21,38], was optimizing a design to obtain higher GOPs, maximize the performance, or reducing power consumption. On the contrary, our focus is to increase the frequency and keep inference engines idle to save dynamic power consumption. The drawback of HARP is that it needs more resources than others, and we justified the demand for more resources by Figure 13. Notably, the reported GOPs in Table 6 exclude the APL computations to make the comparison fair. We did not consider their latency associated with the data transfer time from the sensor to the GPU or FPGA.

Next, we compare our work with the works for near sensor processing. Authors in [39] developed edge-cloud collaborative operation exploiting the visual attention. They implemented a 144-PE array at the sensor with a JPEG encoder and reported efficiency in every layer. The HARP architecture shows more energy efficiency compared to their work. For instance, we need **28.5%** less energy than their first convolutional layer (see Figure 13a). The authors in [40] presented CNN inference architectures embedded on a pixel processor array (PPA) near the vision sensors aiming to execute CNN at the sensor, and it represents the example Figure 1c. They have a high speed up, and the near sensor processing allows them to achieve a 3K frame rate for the MNIST dataset using the LeNet5 model. However, VGG-16 is more complicated than LeNet5.

Based on the performance comparison, HARP demonstrates that it can compete with the other state-of-the-art. However, we have the added benefit of reducing the reported latency and power based on the input image feature.

Overall, pushing the inference computation at the sensor node is challenging for the low area, power, and timing budget. The implementation demonstrates that we can integrate the HARP architecture close to the sensor. Though the area overhead is large, we have seen that HARP lessens the of-chip communication while demands lower on-chip memory. Furthermore, the gradual degradation of image data volume and energy with the increase in irrelevant regions is another promising feature of this architecture. We present necessary results and performance comparisons to validate our claim for achieving high-performance.

## 5. Conclusions

This work presents HARP, an efficient hardware architecture that supports time-critical applications by integrating hierarchical processing directly at the sensor interface. Since the processing begins at the image source, machine vision applications experience high speedup by exploiting its large bandwidth. We inherit attention-oriented processing found in the visual systems in the HARP to prune spatiotemporal redundancies and enable high-level processing with salient regions of an image. Pruning the redundancies does not impact the accuracy; instead, it gives energy-saving up to 45.82% with 35.63% speedup under certain conditions. This dynamic energy-saving is crucial for edge devices like image sensors to maintain SWaP constraints. Furthermore, the deployment of visual attention in the circuit brings area overhead, but the increased throughput overcomes the drawback. Moreover, the pairwise convolution operation in the inference engines also reduces the on-chip memory requirement and lessens the number of communications with off-chip memory. Therefore, the HPRP provides high speedup with comparatively less memory requirement while saves energy based on the image feature and enables high-level processing at the sensor for sensor-level information creation.

## Figures and Tables

**Figure 1 sensors-21-01757-f001:**
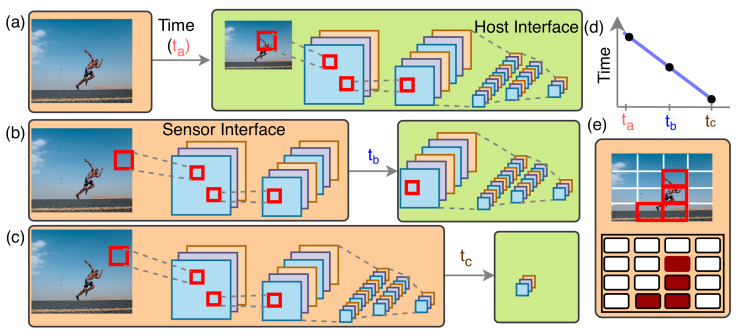
(**a**) Conventional image processing. Sensor is a passive device and continuously sends image to the server or host for processing (**b**) Sensor and host collaborative computation. The vision task is divided between the sensor and host platform (**c**) Full computation on the edge and the sensor becomes active device (**d**) Data propagation delay in these three steps (**e**) Our approach of attention-oriented relevant information processing at the sensor.

**Figure 2 sensors-21-01757-f002:**
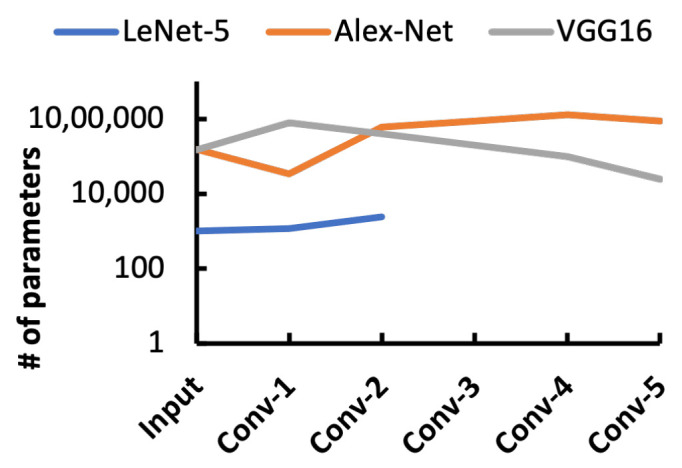
Parameter expansion in different learning models.

**Figure 3 sensors-21-01757-f003:**
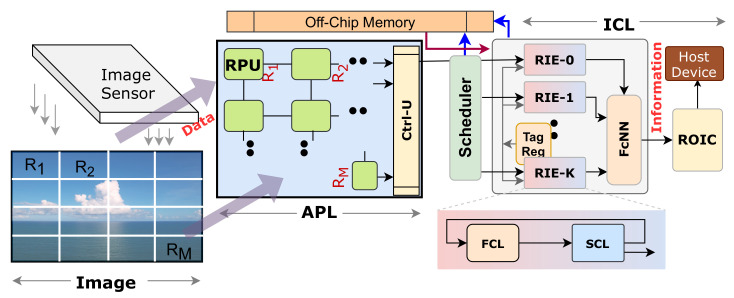
Proposed HARP architecture for CNN integration at the image sensor. The sensor populates the RPUs in parallel. The APL identifies the relevant regions and the ICL computes convolution on the relevant regions.

**Figure 4 sensors-21-01757-f004:**
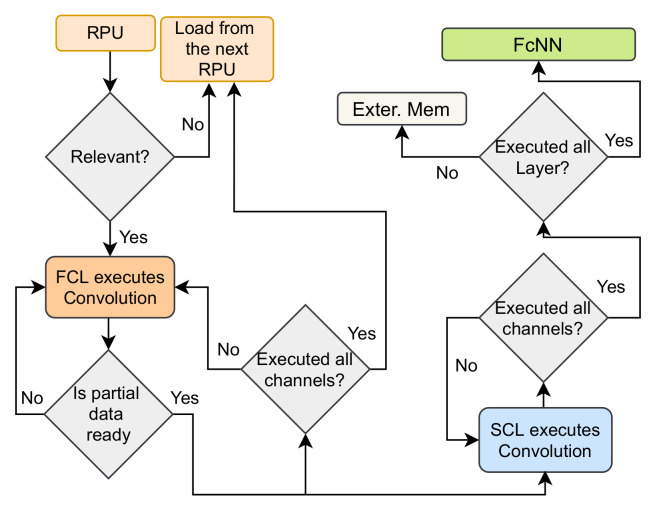
Simplified flowchart of the proposed architecture.

**Figure 5 sensors-21-01757-f005:**
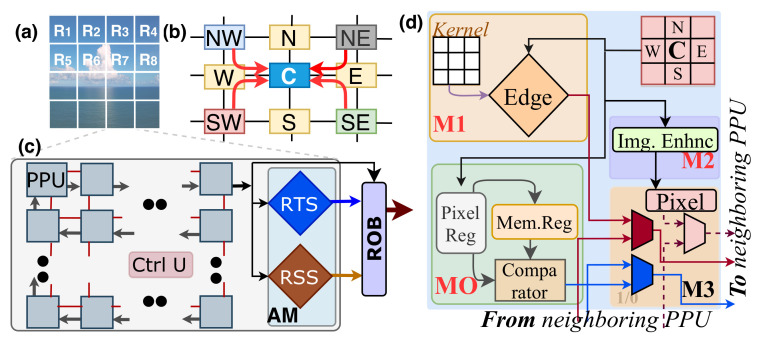
(**a**) Block diagram of an RPU at the Attention-based preprocessing layer (APL) (**b**) Parallel image acquisition process in the sensor to initiate processing (**c**) Block diagram of a RPU (**d**) Block diagram of the PPU.

**Figure 6 sensors-21-01757-f006:**
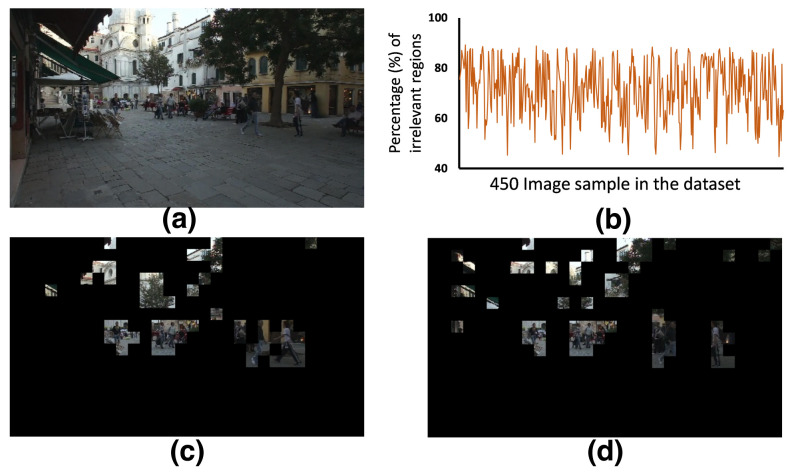
(**a**) An original image in MOT17-SD1 dataset [31] (**b**) Percentage regions with Temporal redundancy in the dataset (**c**,**d**) image regions without redundancies (effective data volume is very low).

**Figure 7 sensors-21-01757-f007:**
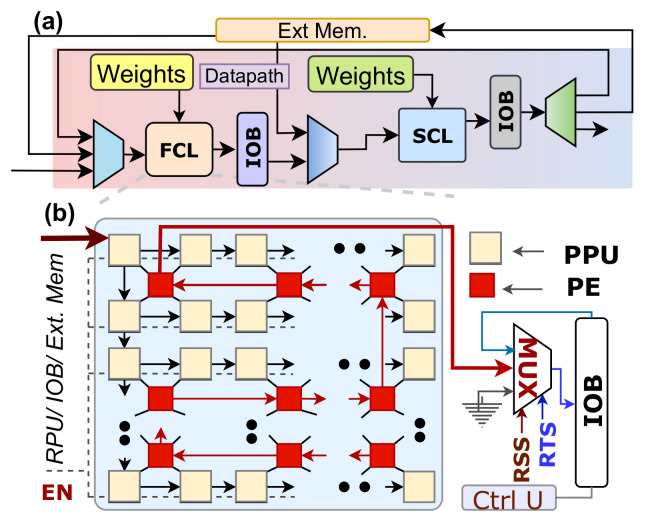
(**a**) Block diagram of an inference engine. There is configurable data-path between FCL and SCL (**b**) Block diagram of the FCL/SCL.

**Figure 8 sensors-21-01757-f008:**
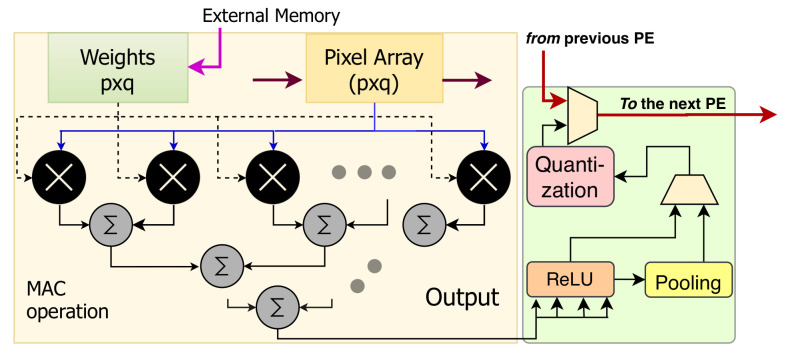
Block diagram of the Processing Unit (PE). It performs MAC operations, rectification, pooling (when required), and quantization.

**Figure 9 sensors-21-01757-f009:**
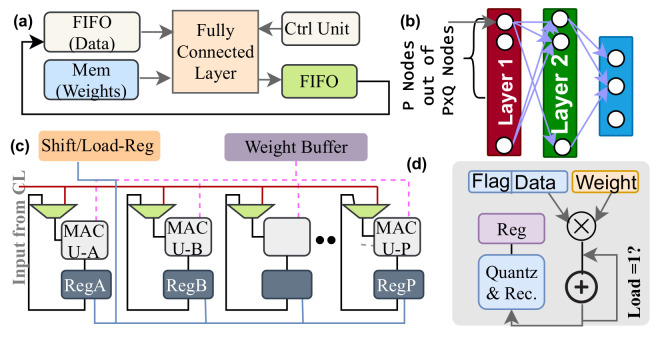
(**a**) Block diagram of the fully connected layer (FcNN) (**b**) An example of FcNN in any learning model (**c**) Functional diagram of the FcNN (**d**) MAC unit in the FcNN.

**Figure 10 sensors-21-01757-f010:**
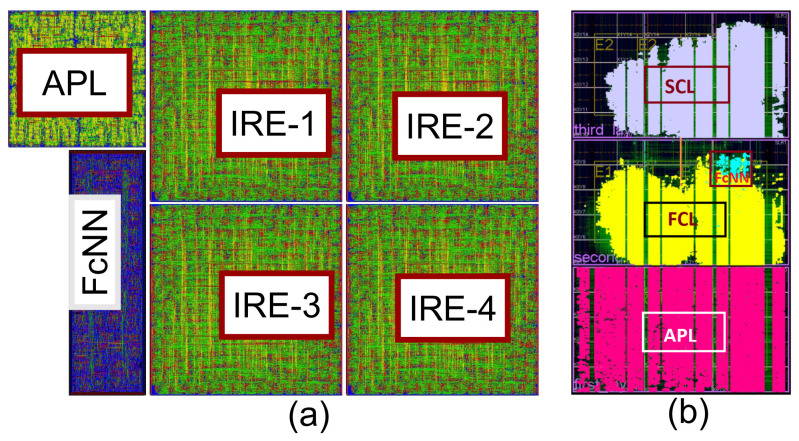
Layout of HARP with four Inference Engines in ASIC using TSMC library at 90 nm technology (**a**) and in the Virtex UltraScale+ FPGA (**b**).

**Figure 11 sensors-21-01757-f011:**
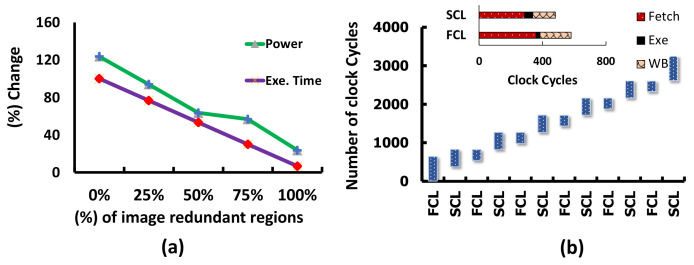
(**a**) Percentage change of power and time with image relevance (**b**) Pipeline processing system for image convolution for one image patch. (Inscribed: Time required to fetch, Exe and WB for different modules).

**Figure 12 sensors-21-01757-f012:**
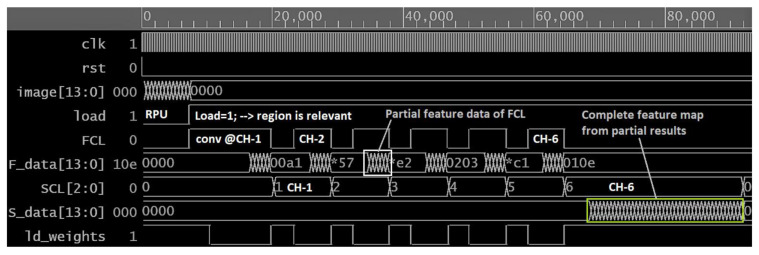
Simulation Result of HARP.

**Figure 13 sensors-21-01757-f013:**
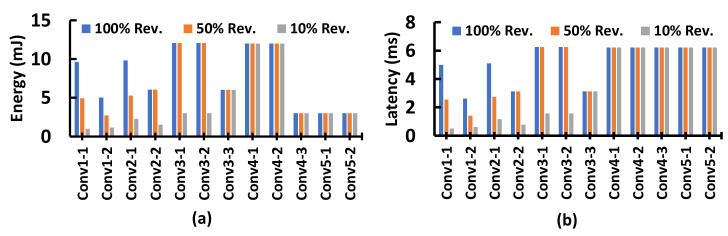
(**a**) Energy consumption at the different stages in the processing hierarchy. (**b**) Time required to perform computation at every stage for the VGG-16 model. The analysis is made under three different 100%, 50%, and 10% relevant regions.

**Table 1 sensors-21-01757-t001:** Impact of temporal and spatial saliency score for next layer processors.

RTS	RSS	Impact on the Next Layer	
		Region Processor	Output from the Region
1	1	Active	Driven by current state
0	1	Inactive	Driven by previous state
1	0	Inactive	Forced to Zero
0	0	Inactive	Forced to Zero

**Table 2 sensors-21-01757-t002:** Impact of Attention Module Threshold (8 ×8 region size).

Threshold Value	≤5	≤10	≤15
Avg. Irrelevant regions (MNIST)	50%	62.5%	68.7%
Avg. Irrelevant regions (Fashion MNIST)	21.8%	29.3%	35.6%
Accuracy Drop (MNIST)	0%	0.9%	5.8%
Accuracy Drop (Fashion MNIST)	0%	0%	1%

**Table 3 sensors-21-01757-t003:** Parameters used in this design.

Parameter/Module	Dimension	Parameter/Module	Dimension
Image Size (M × N)	224 × 224	No. of Regions (M)	196
Region Size with neighboring pixels	18 × 18	Pixels in a region (N)	256
RPU Size	16 × 16	No. of RIE	4
FCL Size	16 × 16	Kernel size (p × q)	3 × 3
SCL Size	14 × 14	PEs in FcNN (P)	100

**Table 4 sensors-21-01757-t004:** UltraScale FPGA Resource Utilization.

	LUT	FF	DSP	BRAM	Power (W)	Fmax
APL	154,404	210,744	0	-	2.044	
RIE	318,016	450,896	2,048	-	7.876	320
FcNN	3,982	3,632	104	-	0.47	MHz
Overall	469,288	663,488	2,100	27	10.68	
Utilization	39.68 (%)	28.06 (%)	30.70 (%)	1.25 (%)	-	

**Table 5 sensors-21-01757-t005:** Layout extracted parameters of the HARP Architecture in ASIC.

Modules	Area (µm${}^{2}$)	Power (mW)				Delay µS	Wire Length µm	>Density (%)	Fmax MHz
		Internal	Switching	Leakage	Total				
APL	950,989	314.6	51.56	6.48	372.68	0.71	3,125,754	71.2	430
IRE	1,830,866	408.1	142.14	10.33	581.4	2.47	6,773,609	68.05	
FCL	996,853	214.5	81.07	7.14	302.7	1.53	3,466,860	69.11	
SCL	697,797	160.5	63.19	5.31	229.05	1.12	260,0145	68.8	
FcNN	70,885.6	20.84	3.70	0.52	25.06	0.01	149,958	69.43	

**Table 6 sensors-21-01757-t006:** Performance Comparison.

	[9]	[19]	[21]	[38]	[20]	[40]	[39]	HARP	
Medium	GPU	Xilinx FPGA	Intel FPGA	Xilinx FPGA	Xilinx FPGA	Cust. Hard.	ASIC	Xilinx FPGA	ASIC
Device	GTX Titan Black	Zynq XC7Z045	Arria10 GX1150	Zynq XC7Z045	Kintex KU060	Image Sensor	28 nm Tech.	Virtex XCVU9P	90 nm Tech.
Model	VGG-16	VGG-16	VGG-16	VGG-16	VGG-16	LeNet5	VGG-16	VGG-16	VGG-16
Precision	32-bits	8-bits	16-bits	16-bits	16-bits	13-bits	16-bits	14-bits	14-bits
Logics/Area	∼	30 K	138 K	218 K	433 K	∼	1.079 mm${}^{2}$	469 K	9.2 mm${}^{2}$
Fmax	∼	167 MHz	200 MHz	100 MHz	200 MHz	∼	∼	320 MHz	430 MHz
Latency	128.62 ms	84.75 ms	43.2 ms	95.48 ms	∼	∼	∼	47.19 ms	34.78 ms
GOPs	∼	135	30.95	57.31	45.07	∼	∼	49.92	58.28
FPS	7.8	11.8	23.14	10.47	∼	3000	∼	21.18	28.75

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
