# Peer review of "HARP: Hierarchical Attention Oriented Region-Based Processing for High-Performance Computation in Vision Sensor"

_sensors, 2021, doi:10.3390/s21051757_

Round 1

Reviewer 1 Report

Dear authors,

First, I would like to congratulate you for the excellent work carried out. Nevertheless, I have some doubts and comments that may be you could clarify.

  1. In the abstract, you claim a 3.68 speed-up compared to state-of the art technology. I would remove this claim since in your performance comparison table (table 5) that speed-up is versus the worst reference [8], which has only 7.8 fps. The gain compared faster approaches is not so good. Moreover it is claimed that your system is up to 45.82% more 14 energy efficiency, but reading the experimental part, that is true in some ideal conditions (when there is a 10% of relevant information). I would rewrite this sentence to be more honest. I would way something like “while in certain conditions saving up to 45% more energy efficiency with the attention-oriented processing”
  2. I would use the comma as thousands separator. In this way, I would change line 19 “10000” –> “10,000”.
  3. Figure 1 caption: I think there is a spelling mistake “sesnor” -> “sensor”.
  4. In the introduction, when talking about “data deluge” (line 52) and redundant Information, I missed some references about some different approaches to solve this data redundancy problem. Let me suggest some of them:
    • Space variant vision: Eric L. Schwartz, Douglas N. Greve, Giorgio Bonmassar, Space-variant active vision: Definition, overview and examples, Neural Networks, Volume 8, Issues 7–8, 1995, Pages 1297-1308, ISSN 0893-6080, https://doi.org/10.1016/0893-6080(95)00092-5.
    • AER sensing: G. Gallego et al., "Event-based Vision: A Survey," in IEEE Transactions on Pattern Analysis and Machine Intelligence, doi: 10.1109/TPAMI.2020.3008413.
    • Selective Change Driven vision: Jose A. Boluda, Pedro Zuccarello, Fernando Pardo and Francisco Vegara. Selective Change Driven Imaging: A Biomimetic Visual Sensing Strategy. Sensors. ISSN: 1424-8220. Vol. 11, Issue 11. November 2011. pp: 11000-11020. DOI: 10.3390/s111111000
  5. In section 1, page (line 64) you point out “the state-of-the art works for salient point extraction involved complex computation like color, etc”. I understood from that sentence that your approach works with grey levels, which could be sufficient if the results are good enough.
  6. Section 2, line 96, you talk about the “drawback in GPU” solved by embedded GPU. Are you talking about power consumption or bandwidth? Please, specify.
  7. In line 110, it appears for the first time MAC, word which is not included in the list of abbreviations, and that is not defined, if I am not wrong until th 9th page, line 288.
  8. Line 137, “saves” -> “save” ?
  9. In my opinion, in each subsection of section 3, explaining each part of the system, you should briefly explain where it is implemented: in the plane sensor, in an external FPGA, in an external board, etc. Otherwise, it is easy to get lost with the infinite amount of modules and abbreviations.
  10. In the 3rd section I got lost in some details. When you talk about “M number of logical regions where each region is a 2D array of N pixels” why you did not say what those values are? I understand that those blocks are hardware blocks and can not be changed. Please give a number.
  11. In figure 3 I think that the word “Schedular” is misspelled and you should write “Scheduler”.
  12. Line 155, RPU meaning should be introduced in the text.
  13. Line 156, Please indicate the number of PPUs N.
  14. About the image acquisition , have you design a CMOS sensor? Have you used reference [1], a 10K FPS, as your sensor? I understand that the visual acquisition has to be integrated together with the PPU processing part. Otherwise, I do not understand how acquired data are transferred.
  15. Line 178 you say “notable change in the window” I understand, as later is confirmed, that you are defining arbitrary thresholds for defining what is notable change. Please, specify here that you are using thresholds.
  16. The AM, although defined in line 156 is not in the List of abbreviations.
  17. Line 188. Please, give a brief explanation of the need of the asynchronous FIFO.
  18. Line 206, a brief explanation about why 8x8 region size should be made. Why this particular size?
  19. In Line 218 you said “It should be noted that finding the most optimal algorithm for calculating image relevance is not a vital contribution of this paper”. I partially disagree, because in my view it is key in your paper to compute the image relevance in order to select the right regions to be processed, and therefore the validity of your hierarchical approach. I simply would delete this sentence.
  20. Line 222, please give any figure about why the ICL is so demanding: number of cycles, processing time (relative or absolute). Just to have an idea of how long it takes.
  21. Line 261 “3.2M byte” -> “3.2 Mbytes” ?
  22. Lines 262-264. Those two consecutive “However” in my view make the text hard to understand. Please, consider rewrite these sentences.
  23. Line 296 “participates” -> “participate” ?
  24. Line 307 “induce” -> “induces” ?
  25. Line 327 “patch” -> “path” ?
  26. Line 336 “FCL” -> FcNN ?
  27. Line 342 “QX ” a factor is missing here?
  28. Line 378 I do not understand “Here, we the region size of 18 x 18 for input image frames of 224 [1]” Please, consider rewrite the sentence.
  29. Line 383 “Implementation Detail” -> “Implementation Details”
  30. Line 384 “The design has prototyped” -> “The design has been prototyped”
  31. Have you actually build the ASIC? I understood that you have prototyped but I do not know if the ASIC has been really built and used in the experimental part.
  32. The whole 4.3 subsection is very detailed and well written. Now some previous doubts, like the asynchronous FIFOs, are clarified.
  33. I see that the weakness of your approach is that uses more resources than others do. This is not a problem in my view, if some other parameters are improved, like energy reduction, while keeping good enough FPS. I would introduce this drawback in the final part of the abstract.
  34. As mentioned at the beginning, in the conclusion part you claim a speedup of 3.68x, which is true compared to a reference [8], a 2014 work. Nevertheless, your FPS are similar to [17]. It is quite evident that this comparison is biased. I see the benefits of your architecture in having a good enough speed, while saving energy, when there is image redundancy, which happens almost always. I would not claim that speed up figure.

Author Response

Dear Respected Reviewer,

We are grateful to you for spending your valuable time reviewing our paper. Your reviews were very constructive, insightful and gave us a proper direction to improve this paper. We have tried our best to follow your guidance and firmly believe that these changes were inevitable. We have attached our response here. The attached letter includes your observations, response, and correction we made in the manuscript based on your review. Your comments, our responses, and the changes we made in the manuscript have black, blue, and brown colors, respectively, for more visibility.

Thank you again for your careful review.

Sincerely,

Pankaj Bhowmik

On behalf of the Authors.

Reviewer 2 Report

This is an interesting work where the authors presented hardware architecture to develop a high-performance image sensor.  The proposed model was prototyped in FPGA and ASIC for integration with a pixel-parallel image sensor. While the presented results are promising, following aspects need to be considerd. 

  1. The conclusion section is not clear, this section needs to be improved by highlighting the actual contribution this paper will mak in the knowledge domain.
  2. there is more recent work reported in the literature which needs to be included: Ghani, A.; See, C.H.; Sudhakaran, V.; Ahmad, J.; Abd-Alhameed, R. Accelerating Retinal Fundus Image Classification Using Artificial Neural Networks (ANNs) and Reconfigurable Hardware (FPGA). Electronics 2019, 8, 1522. https://doi.org/10.3390/electronics8121522
  3. While the FPGA architecture has been presented, there are no simulation waveforms included. How did you verify the design? what is the evidence that design worked as intended? 
  4. How readers can reproduce results? please must include a table with all the parameters used.
  5. An overall flowchart/state machine is needed at the system level to elaborate on the design.

Author Response

Dear Respected Reviewer,

We are grateful to you for spending your valuable time reviewing our paper. Your insightful comments have given a direction to improve this paper. We have provided our best effort to follow your advice and strongly agree that these modifications were necessary for adding more value to this work.

Please find the attachment to reevaluate our work. The attached letter includes your observations, response, and correction we made in the manuscript based on your reviews. Your comments, our responses, and the changes we made in the manuscript have black, blue, and brown colors, respectively, for more visibility.

Thank you again for your careful review.

Sincerely,

Pankaj Bhowmik

On behalf of the Authors.
